# Longitudinal Study on the Antimicrobial Performance of a Polyhexamethylene Biguanide (PHMB)-Treated Textile Fabric in a Hospital Environment

**DOI:** 10.3390/polym15051203

**Published:** 2023-02-27

**Authors:** Sui-Lung Yim, Jessie Wing-Yi Cheung, Iris Yuk-Ching Cheng, Lewis Wai-Hong Ho, Suet-Yee Sandy Szeto, Pinky Chan, Yin-Ling Lam, Chi-Wai Kan

**Affiliations:** 1Avalon SteriTech Limited, Pok Fu Lam, Hong Kong, China; 2Property Management and Supporting Services Department, United Christian Hospital, Hong Kong, China; 3School of Fashion and Textiles, The Hong Kong Polytechnic University, Hung Hom, Hong Kong, China

**Keywords:** PHMB, antimicrobial clothing, antimicrobial medical textile, infection control, contaminants, infection transmission vehicles, disinfection, coronavirus

## Abstract

Healthcare workers in the hospital environment are at risk of infection and body fluids such as saliva, bacterial contamination, oral bacteria, etc. directly or indirectly exacerbate this issue. These bio-contaminants, when adhered to hospital linens and clothing, grow substantially, as conventional textile products provide a favorable medium for bacterial and viral growth, adding to the risk of transmitting infectious diseases in the hospital environment. Textiles with durable antimicrobial properties prevent microbial colonization on their surfaces and help contain the spread of pathogens. This longitudinal study aimed to investigate the antimicrobial performance of PHMB-treated healthcare uniforms during prolonged usage and repetitive laundry cycles in a hospital environment. The PHMB-treated healthcare uniforms displayed non-specific antimicrobial properties and remained efficient (>99% against *S. aureus* and *K. pneumoniae*) after use for 5 months. With the fact that no antimicrobial resistance was reported towards PHMB, the presented PHMB-treated uniform may reduce infection in hospital settings by minimizing the acquisition, retention, and transmission of infectious diseases on textile products.

## 1. Introduction

Infectious diseases pose a significant threat in hospital environments. Pathogens such as methicillin-resistant *Staphylococcus aureus* (MRSA), vancomycin-resistant *Enterococci* (VRE), *Clostridium difficile*, norovirus, coronavirus, and multidrug-resistant Gram-negative rods are known to survive on dry surfaces for extended periods and are difficult to eliminate by simple cleaning and disinfection procedures [1]. Contaminated surfaces contribute significantly to the transmission of healthcare-associated infections (HCAI) in hospital environments and are evaluated through modeling simulations [2], microbiologic studies [3,4], observational studies [5,6,7,8], and interventional studies [9,10,11,12,13]. In particular, the literature reports the plausible mechanisms of transferring HCAI-causing microorganisms from hospital textiles onto skin and other surfaces [14]. Durable hospital-grade antimicrobial textiles could play a crucial part in infection control. They provide an environment that discourages the growth of microorganisms and suppresses microbial colonization. As a result, applying antimicrobial textiles can potentially remove one of the environmental sources of microorganisms. There are numerous antimicrobial treatment methods reported for use in textiles [15]; however, most of them show limited biocidal efficacy. To make matters worse, their antimicrobial properties deteriorate upon abrasion from daily use and repetitive hospital-grade laundry cycles [16]. To overcome the major drawbacks of commercial antimicrobial fabrics, several recent studies evaluated the fabrication of silver(0) [17], copper(II) [18], and Ti_3_C_2_/BiOBr [19] on fabrics via a molecular engineering approach. The stable coordination between hydroxyl groups of cotton fiber and the biocidal metal components minimizes chemical leaching and provides improved antimicrobial durability. They demonstrate promising antimicrobial characteristics for future applications in the healthcare sector. Still, these technologies raise other practical concerns in daily applications. For example, there would be regulatory considerations on novel active biocidal substances, such as Ti_3_C_2_/BiOBr, on treated fabrics, not to mention that impregnated copper(II) can alter the fabric’s original color. As a result, they are not an immediate practical solution in the hospital environment.

Polyhexamethylene biguanide, also known as PHMB, is a polymer containing multiple cationic biguanide-repeating units separated by hydrophobic hexamethylene hydrocarbon chains (Figure 1). When present at a low concentration (<0.3% wt/wt) [20], PHMB is biocidal and widely applied in various products, such as wet wipes [21], cleansing solutions of contact lenses [22], pharmaceutical hygiene products for wound and burn treatment [23,24], and water treatment [25]. It is generally accepted that the positively charged biguanides bind to the negatively charged phosphate group of the bacterial cell wall or virus envelope. This interaction breaks the membrane integrity, leading to cell lysis and subsequent cell death [26]. A recent molecular dynamic study on PHMB revealed another antimicrobial mechanism wherein the PHMB polymer translocates across the membrane bilayer, binds to the DNA, and potentially blocks the DNA replication and repair pathway [27]. Regardless of the antimicrobial mechanisms, PHMB demonstrated non-specific biocidal properties on microorganisms, and no bacterial resistance against PHMB has ever been reported [28]. Attributed to the electrostatic interaction and hydrogen bonding, PHMB shows a strong binding affinity toward cotton fiber [29]. The PHMB–cotton association and simulated washability studies were explained by Kan et al. on a laboratory scale [30]. Utilizing the fact that PHMB is a broad-spectrum biocide and a binding mechanism with cotton, Kan et al. and Avalon SteriTech demonstrated a practical pad–dry–cure treatment method to adhere PHMB on cotton-based fabrics on a mass production scale [31]. To our knowledge, the application of cotton-based antimicrobial uniforms in the hospital environment for infection control is rarely investigated. In this context, a longitudinal study on the antimicrobial performance of PHMB-treated cotton-based healthcare uniforms over a 5-month period against daily wear-and-tear and repetitive laundry cycles was performed and is presented herein.

## 2. Materials and Methods

### 2.1. Antimicrobial Textile Fabric

Cotton/polyester blended fabric is inexpensive but versatile, and can be tailored into breathable, tear-resistant, and abrasion-resistant garment products. Consequently, it has wide applicability in making textile fabrics used in the hospital environment and was selected as the fabric for the composition of the hospital uniform in the present investigation. The fabric consists of 35% cotton and 65% polyester with a weight of 175 g/m^2^. The antimicrobial recipe consists of different ingredients: (i) polyhexamethylene biguanide (PHMB) (15% (weight/volume (*w*/*v*); original purity at least 20%), (ii) poly(ethylene glycol) (PEG) (5% (*w*/*v*); molecular weight: 400, purity >99%), and (iii) binder (8% (*w*/*v*); (polyurethane based, polyurethane content at least 20%). The antimicrobial formulation is applied to the fabric via pad–dry–cure process in which the padding had a wet pick-up of 80%; drying was performed at 90 °C for 5 min and curing was performed at 130 °C for 3 min. This process was applied to the fabric form.

### 2.2. Antibacterial Property

The antibacterial properties of the PHMB-treated textile fabric were tested according to the standard method AATCC TM100-2019 using *S. aureus* (ATCC 6538) and *K. pneumoniae* (ATCC 4352). The standard bacterial solutions were artificially introduced on a treated fabric. After contact for 24 h at 25 °C, the test sample was washed and shaken vigorously with a neutralizing solution. Serial dilution of the neutralizing solution was performed and transferred onto a plate to enumerate the bacteria that remained. The antibacterial rate of the PHMB-treated textile fabric was calculated using the following formula:*Antibacterial Rate* (%) = [(*A* − *B*)/*A*] × 100%(1)
where *A* is the CFU/mL for sample at 0 h and *B* is the CFU/mL for sample after a contact time of 24 h.

### 2.3. Antiviral Property

The antiviral property of the treated textile was evaluated according to the standard ISO 18184:2019 using H1N1 and human coronavirus 229E as viral pathogens. The standard solution containing the virus was added to the treated fabric. The mixture was then inoculated for 2 h at 25 °C. The neutralizing solution was then added to wash out the remaining virus from the specimens. The number of infectious virus particles was then quantified using the Media Tissue Culture Infectious Dose (TCID_50_) assay, and the corresponding antiviral activity value was obtained by the following formula:*Antiviral activity value* (M) = lg(*V_a_*) − lg(*V_c_*)(2)
where lg(*V_a_*) is the common logarithm average of infectivity titer value immediately after inoculation of the control specimen, and lg(*V_c_*) is the common logarithm average of infectivity titer value after 2 h of contact with the PHMB-treated textile fabric. The antiviral activity rate was determined by
*Antiviral Activity Rate* (%) = [(*V_a_* − *V_c_*)/*V_a_*] × 100%(3)
where *V_a_* is the common average of infectivity titer value immediately after inoculation of the control specimen, and *V_c_* is the common average of infectivity titer value after 2 h of contact with the PHMB-treated textile fabric.

### 2.4. Longitudinal Study Design

The study was conducted in an accident and emergency department of a hospital in Hong Kong from December 2021 to May 2022 (with ethical approval from (i) The Hong Kong Polytechnic University (ref: HSEARS20200629003-02) and (ii) Hospital Authority, Hong Kong (ref: KC/KE-21-0154/ER-3)). Before the start of the longitudinal study, the antibacterial properties of the unused PHMB-treated textile fabric were verified, as baseline, by testing against a Gram-positive bacterium, *S. aureus* (ATCC 6538) and a Gram-negative bacterium, *K. pneumoniae* (ATCC 4352). The antiviral properties were verified, as baseline, by testing against an envelope virus H1N1 and human coronavirus 229E. It was found that the unused PHMB-treated textile fabric inactivated, by >99.99%, both *S. aureus* and *K. pneumoniae* (Table 1). In terms of the antiviral property, it inactivated >99% of human coronavirus 229E and >99.99% of H1N1 (Table 2). The PHMB-treated textile fabric was then manufactured into antimicrobial healthcare uniforms as shown in Figure 2.

On the first day of the field study, healthcare workers in the accident and emergency department who enrolled in the study were each provided with an unused PHMB-treated antimicrobial uniform. After the shift work, the uniforms were washed according to the existing washing practice in the hospital laundry, i.e., pre-wash with conventional detergent at ambient temperature to 40 °C for at least 4 min, followed by the main wash procedure for at least 75 °C for 5 min with the presence of 300 ppm hydrogen peroxide and conventional detergent. Subsequently, the uniforms were recycled for use. Three clean uniforms were randomly collected from the hospital laundry on day 30, day 60, day 90, day 120, and day 150, which represent continuous usage for 1 to 5 months, respectively. Each collected uniform (including both the jacket and trousers) was tested against bacteria, *S. aureus* and *K. pneumoniae*, and against an indicative virus, human coronavirus 229E, to evaluate if the antimicrobial properties of the PHMB-treated uniforms show any changes upon continuous usage and repetitive laundry cycles in hospitals.

## 3. Results and Discussion

Overall, 74 healthcare workers from the accident and emergency department of a hospital participated in this longitudinal study. The antibacterial and antiviral properties of the used PHMB-treated healthcare uniforms were evaluated (Table 3).

The reduction values of the unused uniforms were >99% on all tested microorganisms, which demonstrated the superior antimicrobial performance of the PHMB-treated textile fabric and ensured the validity of the testing. It was found that the antibacterial and antiviral efficacy of the jacket did not show any deterioration after 30 days and remained >99%. Furthermore, the antiviral property of the treated uniform fabric did not show any significant deterioration after 150 days, with up to a 95% inactivation rate on coronavirus. Excellent antibacterial and antiviral properties were retained. They were likely attributed to the broad biocidal spectrum of PHMB and the strong affinity of PHMB on cotton fabrics using the specific antimicrobial treatment method and formulations. It is well established that infectious pathogens remain active on textiles for a prolonged period, for example, SARS-CoV-2 can survive on textiles for 48 h [32]. Based on the experimental results of uniform fabric treated with PHMB after 150 days of being deployed for usage in this study, it is believed that the application of such an antimicrobial textile fabric could effectively remove one of the bio-contamination sources in high-risk areas such as hospital environments. The wide adapability of the presented antimicrobial technology on textile fabric was also tested on multiple cotton-based materials on industrial scales, such as a fabric glove and bed linen [31,33]. Therefore, it is expected that the technology and treatment method would be transferrable to other hospital linens, including bedsheets, pillowcases, and curtains for infection control in hospital environments.

With the potential widespread use of PHMB-treated antimicrobial textile fabrics in hospital environments for a prolonged use period, the risk of the development of microbial resistance to PHMB should be considered. In terms of mechanistic considerations, the non-specific inhibition of cell metabolism by PHMB makes resistance development highly unlikely. Indeed, no bacterial resistance to PHMB has ever been reported in clinical samples [23,28]. Nevertheless, it is recommended to develop a continuous monitoring program on the use of PHMB-treated antimicrobial textiles in hospitals.

## 4. Conclusions

Hospital linens are frequently contaminated with pathogenic microorganisms, and this is one of the major sources of bio-contaminations that are attributed to healthcare-associated infections. The present study demonstrated that an antimicrobial TC textile (cotton 35%/polyester 65%) developed by a pad–dry–cure treatment with PHMB could kill >99% of bacteria and inactivate >95% of coronavirus even after >150 days of use and laundry cycles in a hospital environment, evidencing a practical means to develop durable medical-grade antimicrobial uniforms for general hospital staff, including doctors and nurses. The antimcirobial textile could reduce the source of microbial colonization on top of normal cleaning and disinfection protocols, hence providing an additional layer of protection to people in high-risk areas by containing the transmission of pathogens, including SARS-CoV-2, influenza, and antibiotic-resistant bacteria.

## Figures and Tables

**Figure 1 polymers-15-01203-f001:**
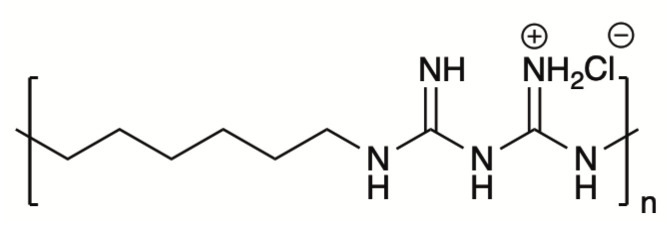
The structure of PHMB.

**Figure 2 polymers-15-01203-f002:**
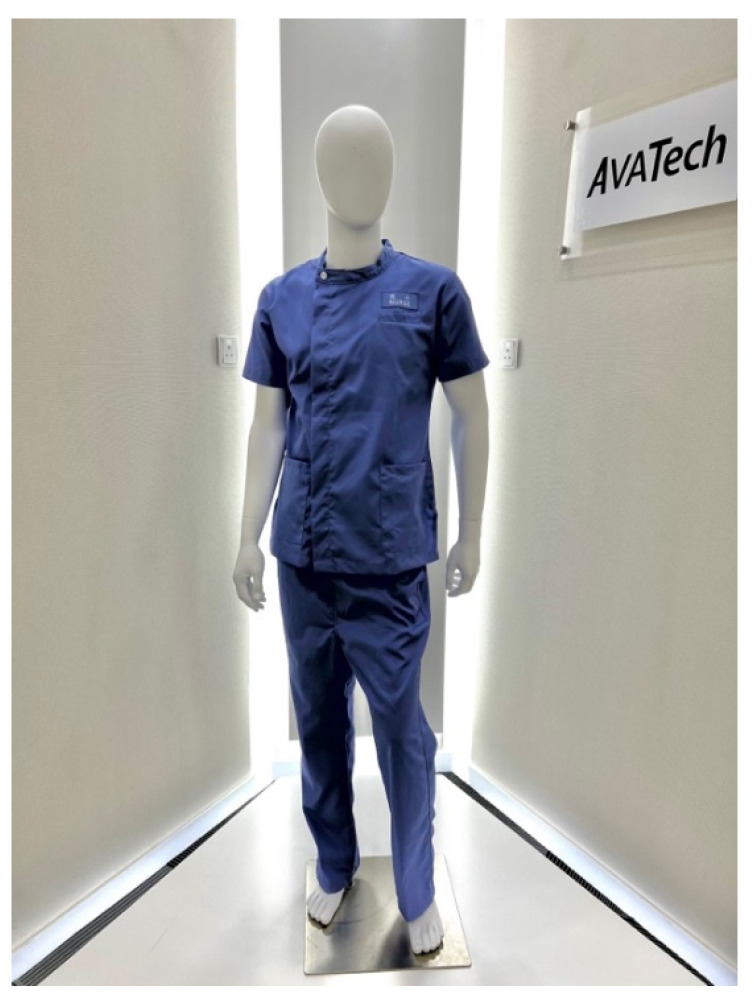
PHMB-treated antimicrobial uniform for hospital use.

**Table 1 polymers-15-01203-t001:** The antibacterial performance (AATCC TM100-2019) of unused PHMB-treated cotton/polyester (35/65) textile fabric for healthcare uniform manufacturing.

Sample	Test Organisms	% Reduction
Control textile fabric	*S. aureus*	N.A.
PHMB-treated textile fabric		>99.99%
Control textile fabric	*K. pneumoniae*	N.A.
PHMB-treated textile fabric		>99.99%

**Table 2 polymers-15-01203-t002:** The antiviral performance (ISO 18184:2019) of unused PHMB-treated cotton/polyester (35/65) textile fabric for healthcare uniform manufacturing.

Sample	Test Virus	Antiviral Activity Value/ Antiviral Activity Rate (%)
Control textile fabric	H1N1	N.A.
PHMB-treated textile fabric		5.11/>99.99%
Control textile fabric	Human coronavirus 229E	N.A.
PHMB-treated textile fabric		2.21/>99%

**Table 3 polymers-15-01203-t003:** Summary of the antimicrobial performance on the used PHMB-treated healthcare uniforms upon different usage periods in a hospital.

Conditions	Antibacterial Rate		Antiviral Activity Rate against Human Coronavirus 229E
	*S. aureus*	*K. pneumoniae*	
Untreated uniforms	N.A.	N.A.	N.A.
Unused uniforms	>99%	>99%	>99%
30 days of use	>99%	>99%	>95%
60 days of use	>99%	>99%	>95%
90 days of use	>99%	>99%	>95%
120 days of use	>99%	>99%	>95%
150 days of use	>99%	>99%	>95%

## Data Availability

Not applicable.

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
