# Peer review of "Longitudinal Study on the Antimicrobial Performance of a Polyhexamethylene Biguanide (PHMB)-Treated Textile Fabric in a Hospital Environment"

_polymers, 2023, doi:10.3390/polym15051203_

Round 1

Reviewer 1 Report

Dear authors,

1. please specify in more detail the material/fibre composition and pretreatments of such materials for treatment with PHMB agent?

2. please explain the common proposed procedure (practise) for washing/cleaning uniforms for hospital environment? Are your proposed temperature and time for treatment sufficient?

3. What about colour fastness and wear comfort with such formulations of antimicrobial agents and the use of hydrogen peroxide in the cleaning process? Have you tested any of the above properties?

4. Do you define the formulation of the cleaning agent used?

5. For which hospital employees can you suggest such treated uniforms? Add a conclusion.

The paper is very interesting, topical and of great attention!

Please answer as best you can, because of the readers.

Author Response

Reviewer 1

1. please specify in more detail the material/fibre composition and pretreatments of such materials for treatment with PHMB agent?

ANS: The fabric material was a blend of cotton-polyester, in a ratio of 35%/65% (wt/wt). Fabric weight was 175 g/m2.

2. please explain the common proposed procedure (practise) for washing/cleaning uniforms for hospital environment? Are your proposed temperature and time for treatment sufficient?

ANS: The procedure of the laundries in current studies were included in the updated main text. The method also represent a common practice in local HK hospital.

3. What about colour fastness and wear comfort with such formulations of antimicrobial agents and the use of hydrogen peroxide in the cleaning process? Have you tested any of the above properties?
ANS: The wear comfort of the PHMB-treated uniform was investigated in questionnaire format. In terms of the material, 50.9% of subjects satisfied and 22.8% of subjects very satisfied the material used in the work clothing. The rest gave neutral feedback and no subject reported extremely dissatisfied comment.

Regarding to handfeel, The result showed 36.8% scored “5” regarding to the handfeel of the work clothing. Scored “4” and scored “6” both ranked by 28.1% subjects, implying that more than total 90% subjects showed positive feedback in this item. Four subjects scored “3” mean that they are neutral about the handfeel of work clothing. No bad feedback for this question received.

The inclusion of hydrogen peroxide in cleaning process is a standard procedure no matter the uniform is conventional or pre-treated with any chemicals. Therefore the colour fastness due to hydrogen peroxide in cleaning process was not investigated in the current studies.

4. Do you define the formulation of the cleaning agent used?
ANS: Conventional detergent was employed in the cleaning procedures.

5. For which hospital employees can you suggest such treated uniforms? Add a conclusion.
ANS: The recommended employees included doctors and nurses in general.

Reviewer 2 Report

The antimicrobial performance of a PHMB-treated textile fabric is demonstrated in this study. This topic is interested to readers especially in a post-pandemic era; however, some flaws should be addressed to warrant the acceptance of the paper.

1. Although the durability of the antimicrobial performance is analyzed via longitudinal study, as a research study in polymer material area, the mechanism for the durability should be carefully analyzed and proved.

2. The washability and abrasive resistance of the PHMB-treated fabrics are suggested to conduct to show the advantages of the making methods.

3. The state-of-the-art textile-based healthcare wearables related to this study are suggested to review to highlight the novelty of this study. Some references are included for your references, such as, doi.org/10.1038/s41565-022-01278-y, doi.org/10.1002/smll.202104448, doi.org/10.1002/adfm.202210351, doi.org/10.1016/j.colsurfa.2022.130506.

 4. The conclusion can be improved to highlight the main findings.

Author Response

Reviewer 2

The antimicrobial performance of a PHMB-treated textile fabric is demonstrated in this study. This topic is interested to readers especially in a post-pandemic era; however, some flaws should be addressed to warrant the acceptance of the paper.

1. Although the durability of the antimicrobial performance is analyzed via longitudinal study, as a research study in polymer material area, the mechanism for the durability should be carefully analyzed and proved.

ANS: The interaction between the biguanide moieties in PHMB vs the carboxylic acid moieties in cotton fibre was investigated in literature by Blackburn. (Blackburn, R.S.; Harvey, A.; Kettle, L.L.; Payne, J.D.; Russell, S.J. Sorption of Poly(hexamethylenebiguanide) on Cellulose Mechanism of Binding and Molecular Recognition. Langmuir 2006, 22, 5636–5644.)

2. The washability and abrasive resistance of the PHMB-treated fabrics are suggested to conduct to show the advantages of the making methods.

ANS: A simulation washability and abrasive resistance of PHMB-treated fabrics was demonstrated in a preliminary study (ref. [30]). The present study demonstrated the washability and abrasion based on actual real-to-use scenarios in hospital.

3. The state-of-the-art textile-based healthcare wearables related to this study are suggested to review to highlight the novelty of this study. Some references are included for your references, such as, doi.org/10.1038/s41565-022-01278-y, doi.org/10.1002/smll.202104448, doi.org/10.1002/adfm.202210351, doi.org/10.1016/j.colsurfa.2022.130506.

ANS: These references were included in the updated introduction of the main text.

4. The conclusion can be improved to highlight the main findings.
ANS: Conclusion was modified to include the summary of the test results.

Round 2

Reviewer 1 Report

Dear Authors,

thank you for your valuable corrections.

Paper could be accept in present form.
